# Anti-Cancer Potential of Transiently Transfected HER2-Specific Human Mixed CAR-T and NK Cell Populations in Experimental Models: Initial Studies on Fucosylated Chondroitin Sulfate Usage for Safer Treatment

**DOI:** 10.3390/biomedicines11092563

**Published:** 2023-09-18

**Authors:** Irina O. Chikileva, Alexandra V. Bruter, Nadezhda A. Persiyantseva, Maria A. Zamkova, Raimonda Ya. Vlasenko, Yuliya I. Dolzhikova, Irina Zh. Shubina, Fedor V. Donenko, Olga V. Lebedinskaya, Darina V. Sokolova, Vadim S. Pokrovsky, Polina O. Fedorova, Nadezhda E. Ustyuzhanina, Natalia Yu. Anisimova, Nikolay E. Nifantiev, Mikhail V. Kiselevskiy

**Affiliations:** 1Research Institute of Experimental Therapy and Diagnostics of Tumor, NN Blokhin National Medical Center of Oncology, 115478 Moscow, Russia; vlasenko2002@bk.ru (R.Y.V.); julius.87@mail.ru (Y.I.D.); irinashubina@mail.ru (I.Z.S.); donenko.f20010@yandex.ru (F.V.D.); d.v.sokolova@gmail.com (D.V.S.); pokrovskiy-vs@rudn.ru (V.S.P.); ppolite@mail.ru (P.O.F.); n_anisimova@list.ru (N.Y.A.); kisele@inbox.ru (M.V.K.); 2Center for Precision Genome Editing and Genetic Technologies for Biomedicine, Institute of Gene Biology, Russian Academy of Sciences, 119334 Moscow, Russia; aleabruter@gmail.com; 3Research Institute of Carcinogenesis, NN Blokhin National Medical Center of Oncology, 115478 Moscow, Russia; nadushka99@gmail.com (N.A.P.); zamkovam@gmail.com (M.A.Z.); 4Department of Histology, Embryology and Cytology, EA Vagner Perm State Medical University, 614000 Perm, Russia; lebedinska@mail.ru; 5Patrice Lumumba Peoples’ Friendship University, 117198 Moscow, Russia; 6Microbiology, Virology and Immunology Department, Sechenov First Moscow State Medical University of the Ministry of Health of the Russian Federation (Sechenov University), 119991 Moscow, Russia; 7II Mechnikov Research Institute of Vaccines and Serums, 105064 Moscow, Russia; 8ND Zelinsky Institute of Organic Chemistry, Russian Academy of Sciences, 119991 Moscow, Russia; takustya@mail.ru

**Keywords:** CAR-lymphocytes, HER2, nucleofection, immune cancer therapy, non-viral transfection, fucosylated chondroitin sulfate

## Abstract

Human epidermal growth factor receptor 2 (HER2) is overexpressed in numerous cancer cell types. Therapeutic antibodies and chimeric antigen receptors (CARs) against HER2 were developed to treat human tumors. The major limitation of anti-HER2 CAR-T lymphocyte therapy is attributable to the low HER2 expression in a wide range of normal tissues. Thus, side effects are caused by CAR lymphocyte “on-target off-tumor” reactions. We aimed to develop safer HER2-targeting CAR-based therapy. CAR constructs against HER2 tumor-associated antigen (TAA) for transient expression were delivered into target T and natural killer (NK) cells by an effective and safe non-viral transfection method via nucleofection, excluding the risk of mutations associated with viral transduction. Different in vitro end-point and real-time assays of the CAR lymphocyte antitumor cytotoxicity and in vivo human HER2-positive tumor xenograft mice model proved potent cytotoxic activity of the generated CAR-T-NK cells. Our data suggest transient expression of anti-HER2 CARs in plasmid vectors by human lymphocytes as a safer treatment for HER2-positive human cancers. We also conducted preliminary investigations to elucidate if fucosylated chondroitin sulfate may be used as a possible agent to decrease excessive cytokine production without negative impact on the CAR lymphocyte antitumor effect.

## 1. Introduction

CAR-T cells are successfully applied for therapy of resistant or recurrent CD19+ leukemia and lymphoma [1]. However, a significant number of patients who were treated with CD19-specific CAR-T cells suffered from long-term B-cell aplasia. Lenti- or retrovirally (α- or γ-retroviral vectors) transduced CAR-T cells constantly express CAR receptors and may persist in the body for a long period and attack not exclusively malignant blasts but all the CD19+ cells, including normal B cells [2].

CAR-T therapy of solid tumors is still facing many more unresolved problems of efficiency and safety than therapy of hematological malignancies [3,4]. CAR-T cells are often inefficient or possess low potential in the treatment of solid tumors because of their inability to reach tumor cells, the variable expression of the targeted antigen, and the immunosuppressive tumor microenvironment. However, safety problems seem to be even more challenging, as tumor-associated antigens (TAAs) of solid tumors, as well as the CD19 antigen on the leukemic blasts, are frequently present in normal tissues. That is why so called “on-target off-tumor” CAR-T cell reactivity may cause life-threatening side effects during CAR-T cell therapy alongside cytokine storm, which is also common for CAR treatment of hematological malignancies [4].

HER2 is a well-defined TAA, which is overexpressed by numerous solid tumors, including breast, ovarian, non-small cell lung, and gastric cancer and osteogenic sarcoma, glioblastoma, etc. [5,6,7]. It makes HER2 an attractable target for treatment of a wide spectrum of different tumors. However, the low but perceptible physiological expression of HER2 in a wide spectrum of benign tissues (epithelial, mesenchymal, and neuronal) should be taken into account when targeting HER2-expressing tumors [8].

A few clones of anti-HER2 murine monoclonal antibodies, such as 4D5 that served as a base for Trastuzumab, were identified and tested for targeted cancer therapy [9,10]. Trastuzumab—a humanized monoclonal antibody that binds to the extracellular domain of HER2—is currently included in standard regimens of treatment of HER2-positive breast cancer [11]. Trastuzumab inhibits tumor cell growth and proliferation. HER2-positive tumor cells are killed by CD16-bearing lymphocytes in the presence of Trastuzumab through antibody-dependent cellular cytotoxicity (ADCC) [11]. However, the efficiency of this targeted therapy is limited due to the primary or acquired resistance of the HER2-positive tumors [12,13]. Thus, development of anti-HER2-CAR-T cells is a challenging task [14].

Therapies based on HER2-specific CAR-T cells have been proven to overcome HER2-positive tumor resistance to monoclonal antibody treatment [15,16,17]. Surprisingly, data by Szöőr Á. et al. indicate that Trastuzumab-based CAR-T cells can successfully combat Trastuzumab-resistant tumors, even though they target the same epitope [17]. The authors suggest that CAR-T cells can penetrate the tumor matrix (which is a barrier for antibodies) and are able to lyse tumor cells with lower HER2 levels. That makes HER2-targeting CAR lymphocytes a suitable cancer immunotherapy alternative in case of primary or acquired tumor resistance to HER2-specific monoclonal antibodies.

Several clinical studies regarding HER2-specific CAR-T cells are listed in the database ClinicalTrials.gov and some of the studies are currently recruiting patients. However, several attempts of HER2/CAR-T cell cancer therapy failed or were withdrawn because of severe side effects [18,19,20]. Nonetheless, other researchers and clinicians tried to fight different types of metastatic cancer with HER2-specific lymphocytes and achieved positive clinical outcomes. The more successful clinical trials used less active types of CAR (second generation versus third generation), lower doses of CAR-T cells, and local administration of the effector antitumor cells [21,22,23,24,25,26,27]. NK cells and their immortalized lines—that do not induce such severe cytokine storm as T cells—are investigated as a promising source for CAR-effector lymphocytes [26,27]. A recent NCT04660929 clinical study that started in 2020 relied on autologous macrophages engineered to contain an anti-HER2 CAR in subjects with HER2-overexpressing solid tumors [28]. Usage of NK cells and macrophages reduces the risk of inducing graft versus host disease (GVHD) and donor CAR cells might be prepared in advance for emergency cases or patients who lack the possibility of producing autologous CAR effectors. The NCT03500991 study involves supplementary modification of T-cells, allowing for their removal in case of negative side effects [29]. The autologous CD4 and CD8 T cells are lentivirally transduced to express a HER2-specific CAR and EGFRt (a truncated form of epidermal growth factor receptor). EGFRt expression facilitates in vivo detection of the administered transduced T cells and, if the CAR-T cells cause unacceptable side effects, can promote elimination of those cells through a cetuximab-induced ADCC response. By targeting several TAAs or co-administration of immune check-point inhibitors, oncolytic viruses are expected to increase treatment efficiency [30,31,32]. Two of these studies intend to exploit effector functions of virus-specific T cells, transduced to express HER2-specific CAR [24,32,33]. All the studies utilize lentiviral transduction, except for NCT04660929, which employs adenovirus for gene modification [28].

Therefore, HER2-specific effector CAR-T cells were shown to be promising in the treatment of a wide spectrum of malignant tumors. However, special care must be taken for the minimization of prominent adverse side effects. Liu X. et al. reviewed different experimental approaches to optimize HER2-specific CAR-T cells [34]. They discuss multiple methods to mitigate HER2-specific CAR-cell toxicity. These approaches include the fine-tuning of CAR affinity so that they attack only tumor cells with high levels of HER2 but do not display significant reactivity against physiologic levels of HER2. Moreover, CAR-T cells may be engineered in such a way that they react only against tumor cells expressing two TAAs.

Different interesting tactics may be useful to “switch on”, “turn off”, or eliminate CAR cells when necessary [34]. One such approach is employed in the NCT03500991 clinical study discussed earlier [29]. A more recent clinical trial, NCT04650451, includes two types of “switches” allowing both to “turn on or off” anti-HER2/CAR-T cells [35,36]. However, all approaches involve supplementary gene modifications, difficult manipulations, or administration of supplementary drugs that make the technology more complicated and expensive. Conceivably, the optimization of CAR-cell regimens or way of administration are the simplest and the least expensive tactics. Interestingly, HER2-specific CAR-T cells were proved to be highly effective even in very small numbers in an experimental model [15].

To study the antitumor efficiency of different CAR constructs in plasmids for transient mammalian expression, we created second and third generation HER2-specific CAR constructs with interchangeable modules. That means the possibility to add or change signaling moieties or even exchange an scFv, which determines CAR specificity. We decided to use plasmids for transient mammalian protein expression to modify human T and NK cells, as this approach helps to reduce possible adverse effects in a very simple way. Supposedly, several courses of administration of lymphocytes with temporal HER2-specific CAR expression may be an effective and safe treatment option for HER2-positive cancer. Compared with viral and RNA-based vectors, plasmids are easier and cheaper to produce, ship, and store and have a much longer shelf life [37]. Furthermore, the modular nature of plasmids also allows for straightforward molecular cloning, making them easy to manipulate and design for therapeutic use. It is well documented now that lentiviral transduction may cause unpredictable host DNA damage with the possibility of malignant transformation of the transduced lymphocytes [38]. In order to avoid such a risk of host DNA damage as a result of viral insertion, we conducted gene modification via a rapid and safe nucleofection method. That makes our approach rather straightforward, easy, and less expensive, making potential CAR-based therapy more affordable. We believe this approach may provide an effective and safe therapy of HER2-positive tumors, as our CAR-effector lymphocytes exert robust though temporal antitumor effects in vitro and in vivo.

Severe side effects of CAR therapy can be managed by different approaches [39]. The transient expression of CAR is required for the reduction of undesirable side effects from CAR therapy. Altogether, lymphodepletion regimens, chemotherapy, or radiotherapy before CAR-lymphocyte infusion result in pancytopenia [40]. CAR-induced cytokine release syndrome and possible on-target off-tumor immune reactions are managed with immune suppressants, such as steroids, anti-IL-6 receptor antibody, or anti-IL-6 antibody [41]. However, immune suppressive treatment during CAR-based therapy inevitably leads to the diminishment of its efficiency. Thus, the search for new routes to improve hemopoiesis depression in CAR-therapy patients is highly required.

Sulfated polysaccharides, such as fucoidans and chondroitin sulfate and especially fucosylated chondroitin sulfates (FucCS), are promising hemostimulating substances [42,43,44,45,46]. This class of polysaccharides stimulated all hematopoietic germs, increasing the number of leucocytes, red blood cells, and platelets in experimental studies, but possessed low toxicity in contrast to colony-stimulating factors. Moreover, the substances revealed moderate immune-stimulating properties and were able to reduce the IL-6 level, which is one of the cytokine release syndrome factors [42,43,44,45,46]. For instance, FucCS was shown to potently decrease IL-6 to a normal level in a mice model of cyclophosphamide-induced immune suppression [45]. At the same time, it led to a restoration of the number of both white blood cells and red blood cells, as well as platelets, to the control levels in blood. Thus, we supposed that FucCS may be used during CAR-based treatment to lower possible cytokine storm without impairing CAR lymphocyte cytotoxic activity and used as a promising hemostimulator for the prevention of hemopoiesis depression after lymphodepletion regimens before CAR therapy. Based on our initial experiments, FucCS does not reduce CAR-lymphocyte antitumor activity and potentially may be useful for these purposes.

## 2. Materials and Methods


**Cell lines, hybridoma, and culture conditions**


SKOV3, SKBR3, MCF7, HELA, K562, and MTP [47] cell lines were obtained from the collections of NN Blokhin National Medical Center of Oncology. They were cultured in Roswell Park Memorial Institute (RPMI) 1640 medium (Gibco, Detroit, MI, USA) supplemented with 10% fetal bovine serum (FBS) (HyClone, Logan, UT, USA), penicillin (50 U/mL), and streptomycin (50 μg/mL) (PanEco, Moscow, Russia). The cell lines and primary cells were maintained in a humidified atmosphere containing 5% CO_2_ at 37 °C. RONC-aH2 hybridoma was obtained from the collections of NN Blokhin National Medical Center of Oncology.


**RNA isolation**


Total RNA was isolated by RNeasy Plus Mini Kit (Qiagen, Germantown, MD, USA) according to the manufacturer’s instructions. Briefly, 2.5 × 10^6^ cells were sedimented at 1500× *g* at room temperature, resuspended in phosphate buffer saline (PBS) and centrifuged once more in the same conditions. After buffer decantation, the cells were lysed in 350 μL of the lysing solution. The lysate was loaded on the column for genome DNA elimination. The columns were centrifuged for 30 s at 8000× *g*. The resulting cleared solution was mixed with 350 μL of 70% ethanol and applied to the column for RNA isolation. The columns were centrifuged for 15 s at 8000× *g*. The columns were washed three times with RW1 and RPE buffer solutions and additional sedimentations at 8000× *g*. RNA was eluted in 50 μL of H_2_O during 1 min of centrifugation at 8000× *g*.


**CAR constructs**


The antibody from the RONC-aH2 hybridoma belongs to the IgG1 group, κ-type of the light chain. Thus, antisense oligonucleotides mIGLK and mIGHG1 complementary to the 5′-regions of mouse constant parts of heavy and light IgG1 κ were synthesized to clone their variable parts.

Synthesis of the 1st cDNA chain on the RNA matrix from the RONC-aH2 hybridoma was conducted with a Mint Kit (Evrogen, Moscow, Russia) according to the manufacturer’s instructions; with one exception, we used specific oligos (mIGLK or mIGHG1) instead of oligo(dT) 3′-primer. This technology allows to link a known sequence PlugOligo adapter to the 5′-end of the 1st cDNA chain, as it is attached to the oligo(dC) sequence made by the Mint reverse transcriptase at the 3′-end of cDNA. Amplification of variable fragments of the light k- and heavy ɣ1-chains was performed with Kapa HiFi polymerase (Kapa Biosystems, Wilmington, South Africa), mIGLK or mIGHG1 primers, and M1 oligonucleotide from the Mint Kit, which was complementary to the attached sequences to the 5′-ends of newly synthesized first cDNA chains. After, the amplification PCR products were separated on 1.5% agarose gel (Axygen) at 6 V/cm. DNA fragment of the proper size was determined based on GeneRuler DNA LadderMix (ThermoFisher Scientific, Waltham, MA, USA) and excised with the gel. The necessary PCR product was isolated from the gel with Wizard^®^ SV Gel and PCR Clean-Up System (Promega) according to the manufacturer’s instructions. Afterwards, A-tails were attached to the fragment by incubation in the presence of Taq-polymerase and dATP at 72 °C. Then, the fragment was cleaned again with Wizard^®^ SV Gel and PCR Clean-Up System (Promega, Madison, WI, USA). The purified fragment was ligated into pTZ57R/T vector (CloneJET™ PCR Cloning Kit, ThermoFisher Scientific, Waltham, MA, USA). After competent *E. coli* cell transformation, the DNA from the derived colonies was analyzed with specific primers’ pairs M1 and mIGLK (or mIGHG1). The positive colonies were grown over night. Their plasmid DNA was isolated with Wizard^®^ Plus SV Minipreps DNA Purification Systems (Promega, Madison, WI, USA) according to the manufacturer’s instructions. The plasmids were subjected to restriction analysis and sequencing. Sequences were analyzed using the Ensembl BLAST online tool and V and J subtypes were determined for both light and heavy chains.

RNA for intracellular parts synthesis was extracted from activated peripheral blood lymphocytes by RNeasy Plus Mini Kit (Qiagen, Germantown, MD, USA). cDNA was synthesized using RevertAid kit (ThermoFisher Scientific, Waltham, MA, USA) and random hexamer primer (OX40, 4-1BB) or specific primers (CD3zeta-iA, CD28-iA). Chosen parts of CD3ζ, CD28, OX40, and 4-1BB were amplified with corresponding S1 and A1 primers, cloned in pTZ57R/T vector, and sequenced. CD8 transmembrane sequence was constructed using 4 oligos: CD8tm-S1, CD8tm-S2, CD8tm-A1, and CD8tm-A2. After that, the fragments were joined together by overlap extension PCR in intracellular domains: 1) transmembrane CD28-CD28-CD3ζ; 2) transmembrane CD28-CD28-OX40-CD3ζ; 3) CD8tm-4-1BB-CD3ζ. In the next round of PCR, a 3′-part of hinge domain with an HpaI restriction site was added to the 5′-end of the intracellular domains and an EcorR1 site was added to the 3′-end of the intracellular domains. Resulting sequences were cloned once more in pTZ57R/T vector and sequenced.

Final construction was assembled in a plasmid containing the BglII-BamHI-HpaI-EcoRI multiple cloning site. Intracellular parts were transferred from pTZ57R/T via HpaI and EcoRI. The chosen flexible linker (GlyGlySerGly)X3 allowed us to add a BamH I restriction site (GGATCC) for the cloning purposes into the linker coding sequence, as it may be composed from GlySer encoding triplets. The resulting linker coding sequence was ggcggctccggcggcggatccggcggcggctccggc. The variable part of the heavy chain was amplified with the addition of the 3′-part of the linker, with BamHI at the 5′-end. After that, it was phosphorylated and digested with BamHI. It was ligated into BamHI- and HpaI-cleaved vectors containing intracellular domains. The variable part of the light chain was amplified with addition of the 5′-part of the linker, BglII restriction site at the 5′-end, and BamHI restriction site at the 3′-end. After that, the variable part of the light chain was transferred to the plasmids already containing the intracellular domain, the hinge, the variable part of the heavy chain, and half of the linker.

CAR constructs were after that cloned into pMax Cloning^TM^ Vector (Lonza, Cologne, Germany) for mammalian expression as is and with IRES-GFP addition. Primers used for cloning in 5′→3′ format (both sense and antisense ones):mIGLK GCTCACTGGATGGTGGGAAGAmIGHG1 CTGGACAGGGATCCAGAGTTCCAPlugOligo adapter -AAGCAGTGGTATCAACGCAGAGTACGGGGGM1 AAGCAGTGGTATCAACGCAGAGThCD3z-iA taaatgcttcatcctgtgtctca—for cDNA synthesishCD3z-S1 GGCCTGCTGGATCCCAAACTChCD3z-A1 GTTAGCGAGGGGGCAGGGhCD28-iA ctatccagagcagtgatattga for cDNA synthesishCD28-S1 AAGCCCTTTTGGGTGCTGGThCD28-A1 TCGCAGCCTATCGCTCCTGAhOX40-S1 AGGGACCAGAGGCTGCChOX40-A1 TCAGATCTTGGCCAGGGTGh4-1BB-S1 CTGTTGTTAAACGGGGCAGAAAGh4-1BB-A1 CAGTTCACATCCTCCTTCCTTCCTTIC1-S1 TATCGCTCCAGAGTGAAGTTCAGCAGGAGCGCIC1-A1 CACTCTGGAGCGATAGGCTGCGAAGTIC2-S1 ATCGCTCCAGGGACCAGAGGCTGCCIC2-A1 TGGTCCCTGGAGCGATAGGCTGCGAAIC3-S2 GCCAAGATCAGAGTGAAGTTCAGCAGGAGCGIC3-A2 CTTCACTCTGATCTTGGCCAGGGTGGAGTCD8tm-S1 ATCTACATCTGGGCGCCCTTGGCCGGCD8tm-S2 GACTTGTGGGGTCCTTCTCCTGTCACTGGTTATCACCCD8tm-A1 CACAAGTCCCGGCCAAGGGCGCCCAGATGTAGATCD8tm-A2 GGTGATAACCAGTGACAGGAGAAGGACCCIC3-S1 CACTGGTTATCACCAAACGGGGCAGAAAGAAAIC3-A1 GAACTTCACTCTCAGTTCACATCCTCCTTCTTCTTCD3z-EcoRI-A taatGAATTCTTAGCGAGGGGGCAGGGCCD28-hinge-S attaGTTAACtcacacatgcccaTTTTGGGTGCTGGTGGTGGTTGGCD8-hinge-S attaGTTAACtcacacatgcccaATCTACATCTGGGCGCCCTTGGaH2-k-S (start codon underlined) attaagatctATGGATTTTCAAGTGCAGATTTTCAGaH2-k-A CGGAGCCGCCCCGTTTTATTTCCAACTTTGTCCCGSLinker-BamHI-A TAATGGATCCGCCGCCGGAGCCGCCGSLinker-BamHI-S taatggatccggcggcggctccggcaH2-h-S cggctccggcATGAAATGCAGCTGGGTCATCttcaH2-h-hinge-A TTGTCACAAGATTTGGGCTCGGCTGAGGAGACGGTGACCG


**PBMC separation and activation**


PBMCs were prepared by Ficoll-Paque (1.073) (Sigma, Roedermark, Germany) density centrifugation of healthy donor blood buffy coat. Afterwards, the cells were left overnight in RPMI 1640 medium (Gibco, Grand Island, NY, USA) supplemented with 10% FBS (HyClone, Logan, UT, USA), penicillin (50 U/mL), and streptomycin (50 μg/mL) (PanEco, Moscow, Russia). Viable cells were counted via trypan blue (PanEco, Moscow, Russia) exclusion assay in a Luna II automated cell counter (Logos biosystems, Anyang, Gyeonggi-do, Republic of Korea). Non-adherent PBMCs were harvested and centrifuged at 1000× *g* at room temperature, diluted to 4 × 10^6^ cells/mL in the same fresh medium, and transferred into vials preincubated with antibodies to human CD3 (0.5 μg/mL) and CD28 (2 μg/mL) (abcam, Waltham, Boston, MA, USA) as follows. Solutions of monoclonal antibodies to human CD3 (0.5 μg/mL) and CD28 (2 μg/mL) (abcam, Waltham, Boston, MA, USA) in sterile PBS were incubated in sterile vials (SPL life Sciences, Pochon, Kyonggi-do, Republic of Korea) for 5 h, with approximately 1.5 mL per 75 cm^2^ area. Afterwards, vials were washed 3 times with sterile 1% BSA (Sigma, Roedermark, Germany) in PBS. PBMCs were activated for 48 h. In some of the tests, FucCS was added (5 μg/mL).


**Electroporation of human-activated PBLs**


Lonza 4D Nucleofector, X-unit (Lonza, Cologne, Germany) was used for non-viral transfection of plasmid DNA into target cells; 20 μL strips or 100 μL cuvettes (Lonza, Cologne, Germany) were used and 5 × 10^6^ cells were electroporated in 100 μL volume and 1 × 10^6^ cells in 20 μL. The electroporation was conducted in the prewarmed-to-room-temperature fully supplemented P3 Primary Cell Nucleofector™ Solution (Lonza, Cologne, Germany). An amount of 5 μg of the vector DNA was added to 100 μL volume and 1 μg of the same DNA was added to 20 μL. Cells were subjected to electroporation through the program for human-stimulated T cells, EO-115. Before starting the nucleofection, vials with OptiMem medium (Gibco, Detroit, Michigan) supplied with 5% of FCS (HyClone, Logan, UT, USA) without addition of any antibiotics were equilibrated in a humidified atmosphere containing 5% CO_2_ at 37 °C. After the nucleofection, the pre-equilibrated medium was immediately added to the cuvettes and the cells were resuspended and transferred into the vials. The cells were incubated in a humidified atmosphere containing 5% CO_2_ at 37 °C in cell culture flasks with filter caps (SPL life sciences, Pochon, Kyonggi-do, Republic of Korea) overnight before tests.


**Flow Cytometry**


Flow Cytometry experiments were conducted with the use of either FACS Cantoo II (BD Biosciences, Franklin Lakes, NJ, USA) or NovoCyte (ACEA Biosciences, San Diego, CA, USA) flow cytometers. Diluted ascitic fluid of mice infected with the RONC-aH2 hybridoma served for analysis of the antibody interaction with different tumor cell lines. FITC-conjugated goat anti-mouse IgG F(ab’)_2_ (Sobrent, Moscow, Russia) were used as secondary reagents to reveal HER2 binding. Phenotypic evaluation of lymphocytes was conducted with the use of the following antibodies: APC-conjugated antibodies to NKG2D, PE-Cy7 to CD56, and PerCPCy5.5 to CD3 (BD Biosciences, Franklin Lakes, NJ, USA). The data were analyzed with Novo-Express 1.5.0 software.


**Analysis of CAR expression**


CAR lymphocytes (from 2 × 10^5^ to 10^6^) were incubated with Biotinylated Human Her2/ErbB2 Protein, His, Avitag™ (AcroBiosystems, Newark, DE, USA) 10 μg/mL in 100 μL of PBS (PanEco, Moscow, Russia) with 1% BSA (Sigma, Darmstadt, Germany) for 30 min in a fridge. Afterwards, cells were washed 3 times with PBS with centrifugations at 1000× *g*. Further, they were incubated for 30 min with streptavidin-PE (Abcam, Waltham, MA, USA) 1μg/mL in 100 μL of PBS (PanEco, Moscow, Russia) with 1% BSA (Sigma, Darmstadt, Germany) for 30 min in a fridge. Afterwards, cells were washed 3 times with PBS with centrifugations at 1000× *g*. The cells were diluted in 400 μL of PBS and analyzed by fluorescent flow cytometry.

**Standard MTT-assay.** For this test, 5 × 10^3^ MTP cells were seeded into a 96-well flat bottom cell culture plate (Nunc, Roskilde, Sjelland, Denmark) and cultured overnight. The next day, 5 × 10^4^ CAR T/NK cells (10× excess) or the same quantity of activated T/NK cells were added. Total incubation volume was 200 μL. Cells were incubated from 3 to 24 h. After the incubation period, 20 μL of MTT (3-(4,5-dimethylthiazol-2-yl)-2,5-diphenyltetrazolium bromide) (Sigma, Roedermark, Germany) was added. The mixes were incubated for 1.5 h longer until purple insoluble formazan appeared in the tumor cells. Afterwards, all the non-adherent cells were removed and the wells were thoroughly washed with PBS. Thus, only adherent SKBR3 cells remained in the wells. Then, 100 μL of dimethyl sulfoxide (DMSO) (PanEco, Moscow, Russia) was added to each well. The formazan was dissolved during half-an-hour of incubation of the plates on the orbital shaker. Afterwards, tumor-cell viability was determined based on adsorption at 540 nm with a microplate reader (BMG LABTECH, Ortenberg, Germany). The control wells contained only tumor cells and the medium. In some of the tests, FucCS (5 μg/mL) was added. Cytotoxicity (CT) was determined as follows:CT = (1 − A540experiment/A540control) × 100%


**CAR cell cytotoxicity real-time determination with XCELLigence instrument**


For real-time analysis of the CAR-T/NK cells’ cytotoxicity towards SKBR3 and SKOV3 cancer cells with the XCELLigence instrument (ACEA Biosciences, San Diego, CA, USA), 5 × 10^3^ tumor cells were seeded into the E-Plate 16 strips (ACEA Biosciences, Hangzhou, Zhejiang Province, China) and cultured for 4 h. Afterwards, 5 × 10^4^ CAR T/NK cells (10× excess) or the same quantity of activated T/NK cells were added. The data (cell index value) were collected every 15 min for 72 h.


**Real-time observation of the CD28-OX-40-HER2-specific CAR-T/NK cytotoxicity against SKOV3 ovary cancer cell line with an automated microscope**


HER2-positive SKOV3 ovarian cancer cells were labeled with calcein-AM (Invitrogen, Carlsbad, CA, USA) (non-fluorescent calcein AM is converted to green fluorescent calcein in live cells). An amount of 20 μL of the 1 mM calcein AM stock solution was transferred to the 10 mL of serum-free Hanks Balanced Salt Solution (HBSS) (PanEco, Moscow, Russia). SKOV3 cells (8 × 10^5^) were detached from 25 cm^2^ plastic flasks (Nunclon, Waltham, MA, USA) with EDTA incubation at 37 °C, pelleted by centrifugation at 1000× *g* for 5 min, and washed once in a serum-free HBSS (PanEco, Moscow, Russia). Then, 1 mL of the staining solution was directly added to the SKOV3 cells. The cells were incubated at 37 °C for 30 min. Afterwards, the cells were pelleted by centrifugation and the staining solution was decanted. The cells were dissolved in RPMI1640 medium (PanEco, Moscow, Russia) with 10% of FCS (HyClone, Logan, UT, USA) at 2 × 10^5^. Then, 1 mL samples of the suspension were transferred to a 24-well culture plate (Nunclon, Waltham, MA, USA). The SKOV3 cells were let to adhere to plastic for 4 h. The CD28-OX-40-HER2-specific CAR-T cells were added to the labeled adherent SKOV3 cells in fivefold excess (10^6^). The shots were taken by the automated microscope Lionheart FX (BioTek, Agilent, Winooski, VT, USA), equipped with a thermostated CO_2_-chamber, every 1 h for 72 h at 200 magnification in the channel corresponding to the green fluorescent calcein.


**In vivo xenograft tumor model and antitumor treatments**


Protocols for the animal study were approved by the Institutional Ethical Animal Care and Use Committee of NN Blokhin National Medical Research Center of Oncology. All animal experiments were performed in accordance with the relevant guidelines and regulations. The ovarian cancer SKOV3 cells (10^7^) were implanted subcutaneously in 0.1 mL PBS with Matrigel (1:1, Corning, Bedford, MA, USA) into the flanks of Balb/c female nude mice (5-week old) on day 0 simultaneously with the equal quantity of 4-1BB-CAR-T/NK cells or CD28-OX40 CAR-T/NK cells. Thus, there were 3 groups (6 mice per group): the untreated control group and 4-1BB-CAR- and CD28-OX40 CAR-treated groups. During antitumor treatment, tumor dimension was measured using a digital caliper. Tumor volume was calculated by the formula V = π/6*L*W*H, where L is length, W is width, and H is height of tumor. All the mice were euthanized on day 41 and the tumors were dissected out. The experiment was repeated three times. Efficiency assessment was carried out using the TGI (tumor growth inhibition, %) formula: [(V_c_ − V_t_)/V_c_] × 100, where V_c_ is the mean tumor size of the vehicle control group and V_t_ is the mean tumor size of the treated group.


**Azure and Eosin staining of xenograft tumors**


After 40 days from the beginning of the experiment, xenograft tumors were dissected. Dissected tumors with surrounding tissues were fixed and embedded. Serial histological sections 5 µm thick were stained with azure and eosin (PanEco, Moscow, Russia).


**Microscopy**


Microscopic examinations were performed with Axioplan 2 imaging and Axiovert 40 CFL microscopes (Zeiss, Oberkochen, Baden-Württemberg, Germany). Microphotographs were taken with AxioVision microscopy system (Zeiss, Oberkochen, Baden-Württemberg, Germany).


**Statistical analysis**


The data were reported as means ± SD. Data were analyzed using Statistica 10. An ANOVA test with Tukey HSD was used to evaluate differences. *p*-values < 0.05 were considered significant.


**Fucosylated chondroitin sulfate (FucCS)**


FucCS from *Cucumaria japonica* [48] was used in this study. This product was kindly provided by Professor A.I. Usov (N.D. Zelinsky Institute of Organic Chemistry, Russian Academy of Sciences).

## 3. Results

### 3.1. Evaluation of RONC-aH2 Antibody Interaction with Tumor Cells

We assessed interaction of RONC-aH2 antibodies with several tumor cell lines. SKOV3 (human ovarian cancer) and SKBR3 (human breast cancer) express high levels of HER2 antigen. MCF7 (human breast cancer) expresses lower levels of HER2. RONC-H2 antibodies strongly interacted with SKOV3 and SKBR3 cell lines and to a lesser extent with the MCF7 cell line (Table 1). MTP melanoma [47] also interacted with RONC-aH2 antibodies, which is indicative of HER2 expression. HELA cervical carcinoma cells and the K-562 myelogenous leukemia cell line did not bind RONC-aH2 antibodies. Evidently, the tested cell lines SKOV3, SKBR3, MCF7, and MTP express different levels of HER2 antigen reacting with RONC-aH2 antibodies on the surface of almost all the cells.

### 3.2. Construction of HER2-Specific CARs

We created a universal cassette with a possibility to change antigen-recognizing domains and to use different combinations of intracellular signaling moieties. The stable sequences were the linker and spacer regions. In order to generate different CAR constructs, techniques of gene fragment splicing by multistep overlap extension PCR (OE-PCR) and ligation of specific exchangeable fragments into introduced restriction sites were chosen. Restriction enzyme sites were introduced to easily make different constructs if required. The scFv of the CAR constructs consisted of the light and heavy chains from the antigen-binding fragments (Fabs) of the IgG from the RONC-aH2 antibodies specific to HER2. Variable parts of the light and heavy chains of the antibodies were mounted by RT-PCR of RNA derived from the hybridoma and sequenced. Sequencing of three positive clones with variable parts of the light κ-chain of the RONC-aH2 antibodies revealed identical sequences of igkv4-63 V- and igkj4 J-segment types. Sequences of three positive clones with the variable part of the ɣ1-chain were identical as well and corresponded to ighv14-4 V and ighj4 J types. A flexible linker, which consisted of triplicated sequence GlyGlySerGly, was chosen to connect the light and heavy chains of the Fabs. We also preserved 22 amino acid signal peptides of light chain for CAR cell surface translocation [49]. The minimal sequence of the IgG1 hinge (GluProLysSerCysAspLysThrHisThrCysPro) linked the scFv to a transmembrane domain. This article presents second and third generation anti-HER2 CAR constructs with the stable receptor region and varying transmembrane and signaling sites. The third generation OX40-CAR includes three activating domains: OX40 signaling moiety added between CD28 transmembrane and signaling regions and CD3ζ signaling domain (Figure 1). The 4-1BB CAR has two activating moieties. A 4-1BB signaling region was introduced between CD8 transmembrane (non-signaling) and CD3ζ signaling domains (Figure 1).

### 3.3. The CAR Construct Expression

Expression of the CAR construct in lymphocytes after electroporation was evaluated on the next day, 24 h after electroporation by different means. Initially, it was evaluated on the base of concomitant expression of the reporter green fluorescent protein (GFP) in the cells transfected with the tandem CD28-OX40-CAR-IRES-GFP construct. The GFP gene was encoded in the long single mRNA product after the CAR gene, separated by an internal ribosomal entry site (IRES). A similar approach was employed by Li H. et al., who evaluated percentages of the T-cells transduced by their CAR construct in a similar way [16]. They registered expression of the third generation CAR via fluorescence of Zs-Green1 placed after IRES in a retroviral vector. In our experiments, 20–80% of activated PBLs from blood samples of different human volunteers expressed GFP after electroporation with the CAR-IRES-GFP construct, with the average being about 30% (Figure 2).

Activated T-cells are easier transfected with CAR constructs. Cytokines secreted by T cells are crucial for their successful activation. As FucCS was found to reduce the IL-6 level [43], we were interested to evaluate its influence on the nucleofection efficiency of the activated lymphocytes. In our preliminary experiments, the addition of FucCS during lymphocyte activation did not impair efficiency of their further transfection and even increased it (24% of GFP-positive cells without FucCS versus 45% in its presence). Thus, activation in the presence of FucCS proved to be advantageous for further gene modification.

In another set of experiments, direct binding of biotin-modified HER2 to CAR lymphocytes was determined (Figure 3). Interestingly, HER2 binding and GFP expression coincided only in some of the cells. Thus, determination of solely GFP expression by tandem CAR-GFP gene constructs is not adequate for proper CAR evaluation on the cell surface.

### 3.4. Electroporation Efficiency and Transfected Gene Expression Stability in Different Lymphocyte Subpopulations

We determined the spectrum of the lymphocytes that underwent genetic modification by electroporation by applying pMax-GFP positive control vector and a panel of fluorescent monoclonal antibodies against T-cell and NK-cell antigens (CD3, CD56, and activating NK-receptor NKG2D). pMax-GFP nucleofection yields high percentages of very bright GFP-positive lymphocytes; this facilitates identification of the electroporated lymphocytes even within minor subpopulations such as NK and NKT cells. Our data indicate that, alongside T cells, NK and NKT cell subtypes were efficiently transfected via nucleofection (Figure 4). Interestingly, GFP expression remained at a rather high level throughout 96 h of incubation in all the lymphocyte subpopulations (Figure 4b).

However, NK and NKT cells seemed to suffer from the electroporation procedure to a higher degree than the T cells, as their proportions significantly decreased after electroporation (Figure 5a,b). It seems that their subpopulations slightly restored after 48–72 h of incubation (Figure 5a), but total lymphocyte numbers may decline during incubation.

Some of the successfully electroporated lymphocytes retain or even gain NKG2D expression after incubation (Figure 6). NKG2D is an activating immune receptor expressed by NK and effector T cells, which is important for tumor cell recognition by NK cells [50].

Thus, the produced CAR lymphocyte preparations were obviously composed of different lymphocyte subpopulations, including innate effector NK cells and NKT cells, which were shown to play an important role in cancer control. NKG2D expression may be also favorable for the anticancer action of our CAR-T/NK cell preparations.

### 3.5. Cytotoxicity of CD28-OX-40-CAR- and 4-1BB-CAR-T/NK Cells towards Selected Tumor Cell Lines

SKOV3 and SKBR3 cell lines with the highest HER2 expression were selected for evaluation of tumor lytic activity of the CAR-T/NK cells. Different methods were used to study their antitumor activity in vitro. Microscopic examination was performed after 4 h co-incubation. It revealed active clustering of CD28-OX40-CAR-T/NK with SKBR3 cells (Figure 7c). In contrast, control activated lymphocytes did not huddle with the tumor cells (Figure 7b). Real-time cell analysis with the XCELLigence instrument revealed interesting details of 4-1BB- and CD28-OX-40-CAR-T/NK cell antitumor activity. The xCELLigence RTCA label-free technology monitors cell numbers by changes in impedance measured through electrodes embedded in proprietary E-Plates. When seeded alone, target adherent cancer cell proliferation rate is registered as an increase in the impedance-related cell index (CI). Effector non-adherent immune cells produce a small baseline level signal due to the absence of tight surface adhesion. When effector lymphocytes are added to adherent target cells, their cytolytic activity causes tumor cells to round up and detach, consequently reducing CI value. Both types of the CAR-T/NK cell acted potently against SKBR3 cancer cells and their activity significantly exceeded non-specific lytic cytotoxicity of activated lymphocytes (Figure 8a).

The death of cancer cells steadily increased throughout the time of incubation, based on CI value decrease. However, the CD28-OX-40-CAR-T/NK proved more efficient than the 4-1BB-CAR-T/NK cells as their cytotoxic activity was much more prominent during roughly the first 25 h of incubation (Figure 8a). Furthermore, 4-1BB-CAR-T/NK cells were shown to be inefficient against SKOV-3 ovary cancer by the real-time analysis (Figure 8b). Their cytotoxicity towards the tumor cell line was largely the same and even lower than the activated lymphocytes. However, a significant anticancer advantage of the CD28-OX-40-CAR-T/NK cells over activated T/NK cells was evident at most of the time points (Figure 8b) and remained so until the end of the observation. Thus, the real-time data proved the third generation HER2-specific CD28-OX-40-CAR-T/NK cells superior in anticancer action compared with the second generation 4-1BB-CAR-T/NK cells.

Encouragingly, both lines of the cancer cells were not able to restart their outgrowth during the entire co-culture period (72 h) with the CD28-OX-40-CAR-T/NK cells. This demonstrated high CAR-T/NK cell activity during this period or even more despite the transient character of the CAR expression. A precise time limit for the CAR expression and activity still remains to be elucidated. In further research, we also intend to sort effector cells and test them individually.

The high antitumor activity of the CD28-OX-40-CAR-T/NK cells has also been demonstrated in a real-time observation by the automated microscope Lionheart FX, equipped with a thermostated CO_2_-chamber. HER2-positive SKOV3 ovarian cancer cells were labeled with the fluorescent dye calcein. The CD28-OX-40-HER2-specific CAR-T/NK cells were added to the labeled adherent SKOV3 cells in fivefold excess. The video file (Appendix A) obtained during 24 h of observation demonstrates fast aggregation of the CAR lymphocytes with the labeled tumor cells. After the interaction with the CAR-T/NK cells, SKOV3 cells started to detach from the plastic, became round-shaped, and seemed to disappear in the clusters of the CAR lymphocytes.

Consistently, with the strong interaction of the RONC-aH2 antibodies with melanoma MTP, the CD28-OX-40-HER2-specific CAR-T/NK cells were able to effectively lyse the HER2-positive melanoma cell line (Figure 9). Strikingly, activated lymphocytes were not able to lyse this tumor cell line.

Cytotoxic activity of CAR-T/NK cells towards SKBR3 cancer cells remained at the same level with the addition of FucCS (81 ± 10% without it versus 73 ± 10% in its presence) (Figure 10). Thus, the polysaccharide under investigation does not negatively influence the antitumor potential of the CAR lymphocytes, at least in vitro.

Thus, the generated CAR-T/NK cells revealed their cytotoxic activity against HER2-positive breast (SKBR3), ovary (SKOV3), and melanoma (MTP) cell lines. FucCS did not impair CAR lymphocyte cytotoxic activity.

### 3.6. In Vivo Activity of the CAR-T/NK Cells in a Murine Model

We tested antitumor activity of the derived CD28-OX-40- and 4-1BB-CAR-T/NK cells in Balb/c nude mice with xenografts of human ovarian cancer SKOV3. Both types of the CAR-T/NK cells showed prominent anticancer activity. Interestingly, stable tumor growth inhibition was monitored from the 15th day until the end of the observation (40 days) (Figure 11).

On the 40th day, the animals were sacrificed and the tumors were dissected, sectioned, and stained with azure and eosin (Figure 12). We observed a marked difference between samples obtained from the treated and untreated animals. In the untreated animals, tumors were represented by tumor cells that formed clusters separated by a well-developed stroma (Figure 12a,b). The stroma was represented by wide interlayers of connective tissue containing an abundance of thin-walled vessels of different diameters. Small focal lymphoid infiltrates were localized in the stroma. There were multiple tumor cells of different sizes or oval or polygonal. The number of tumor cells definitely diminished after CAR lymphocyte injections (Figure 12c,d). Small numbers of tumor cells were diffusely dispersed. Diffuse and focal infiltration of lymphocytes was obvious. Connective-tissue stroma was much less pronounced and represented by more gentle and thin burdens. The number of blood vessels significantly reduced. There were apoptotic bodies that formed small clusters. In contrast with the control, there was evident exudate (or extracellular matrix), which filled the free space between the few diffuse tumor cells. Thus, the micrographs proved high antitumor efficiency of the CAR-T/NK cells.

## 4. Discussion

We developed HER2-specific CARs on the base of RONC-aH2 antibodies that bind HER2-positive tumors such as breast cancer SKBR3 and ovarian cancer SKOV3. Furthermore, we found that antibodies interacted with the melanoma cell line MTP [47]. It was determined by other researchers that melanomas may express the HER2 antigen; thus, HER2 may also be a promising target for the treatment of melanomas [51]. Strobel S.B. et al. observed HER2 overexpression in 3% of human primary melanomas [52]. However, all of the melanoma cell lines they tested uniformly expressed HER2. In line with the data above, anti-HER2/CAR-T cells were found to efficiently eradicate uveal and cutaneous human melanoma in a preclinical study [53].

We constructed second and third generation CARs. First generation CARs contained only CD3ζ stimulatory moieties [54]. However, their activating potential was not optimal. Pule et al. improved the CAR construct by exchanging CD3ζ transmembrane domain with CD28 analogous and adding CD28 costimulatory domain to CD3ζ [55]. Notably, CD28 costimulation increased IL-2-independent proliferation and enhanced the resistance of CAR-T cells to T regulatory cells [54]. Third generation CARs included three activating domains. Other than CD28, other domains such as 4-1BB and OX40 that additionally favor survival and activation of CAR-T cells were added [56,57]. Certain data indicated that second generation CAR-T cells based on CD28 were highly susceptible to exhaustion [58]. Thus, our second generation CAR was based on a more favorable 4-1BB domain in addition to CD3ζ. The third generation CD28-OX-40-CAR-T/NK cells were found more effective in vitro studies than 4-1BB-CAR-T/NK cells; however, their activity was very similar in the SKOV3 mouse xenograft model. Thus, the relative anticancer potential of our CAR constructs should be examined further.

Efficiency of the CAR constructs’ transfection in our experiments was rather high and comparable to results reached by other teams either by retroviral transduction [16] or nucleofection [59]. We did not thoroughly investigate the time-line of the CAR expression in our setting. However, another team observed stable transfected gene expression for 10 days from the similar non-integrating plasmids after electroporation into NK cells [60]. NK cells are an attractive aim for gene modification as they possess high innate antitumor potential due to their ability to recognize and kill malignantly transformed cells. Moreover, CAR-modified NK cells cannot induce graft-versus-host reactions as CAR-T cells can [61,62]. Thus, allogenic CAR-NK cells may be profitable for cancer treatment. However, NK cells proved to be highly resistant to viral gene modification or chemical transfection. Lentiviral and retroviral transduction causes substantial NK cell apoptosis and the low production of genetically engineered NK cells [63,64]. NK cells express high levels of receptors for pathogen-associated molecular patterns, which enables a heightened recognition of viruses and the suppression of NK transduction [65]. Our data, and other authors’ results, demonstrate that electroporation is an effective method of NK-cell genetic modification applying plasmids, RNA, or ribonucleoproteins [66].

The experimental setting in our research seems very similar to the method described by Ingegnere T. et al. [60], who also applied electroporation of CARs in plasmid vectors for temporary expression. Lonza 4D nucleofector yielded very efficient transfection of the same pMax-GFP plasmid into NK cells (65% on the average) with the Neon Transfection System (up to 50%) (Thermo Fisher Scientific, Waltham, MA, USA) used by Ingegnere T. et al. [60]. Importantly, this method offers more prolonged transient gene expression than RNA electroporation; this may promote more durable anticancer action of the transfected CAR lymphocytes.

Malignant transformation or viral infection of the cells induce surface NKG2D ligands (MICA and MICB) expression that makes cancer cells susceptible to immune destruction. Therefore, NKG2D plays an important role in anticancer lymphocyte action and its expression is highly favorable on CAR lymphocyte surfaces. Similar to our data, Ingegnere T. et al. found stable expression of NKG2D and other activating NK receptors in NK cells after electroporation [60]. The nucleofection of CAR constructs into primary human NK cells is an effective way to generate CAR lymphocytes with valuable innate antitumor receptors. Importantly, CAR-transfected NK cells may be prepared in advance as an off-the-shelf anticancer cellular product [67]. The generated biomedical cell product anti-HER2-CAR-T/NK cells were found safe in a course of intraperitoneal injection at a human equivalent therapeutic dose in male and female ICR mice [68].

We suggest the transient expression of CARs in electroporated plasmids as an effective and simple method to increase safety of CAR-based therapy. The electroporation of plasmid vectors significantly reduces the risk of insertional mutagenesis compared with lenti- or retroviral transduction. Moreover, electroporation is cheaper and does not require such complex preparation or dedicated facilities as viral transduction [60]. In our opinion, this approach may help to lower safety problems arising from on-target off-tumor CAR-lymphocyte reactions [69]. Unstable CAR expression is even desirable in a case when such dangerous anti-HER2 reactions against benign tissues are inevitable. Transient CAR expression may be achieved via RNA or plasmid transfection. However, plasmids are more stable than RNA and express CAR products for longer periods than RNA. Several courses of “transient” CAR lymphocyte administration are desirable for prolonged antitumor effects. It is difficult, or even impossible, to obtain enough autologous lymphocytes from cancer patients for several courses of such CAR therapy. However, allogenic cytokine-induced killer cells have been shown to be safe in adoptive cancer therapy [70,71,72]; they do not induce graft versus host disease and may be adapted as a source for CAR lymphocytes.

We have shown that FucCS does not impair cytotoxic activity of the CAR lymphocytes in vitro. Moreover, lymphocytes were properly activated in its presence and were successfully electroporated with CAR constructs. As the substance was shown earlier to downregulate the IL-6 level and stimulate hematopoiesis in vivo [43], we suppose that synthetic analogs of the used FucCS compatible with GMP manufacturing rules may be used during CAR-based therapy in order to prevent its adverse side effects. The most appropriate stages of its usage are still to be elucidated. First of all, it seems that analogs of FucCS might be administered together with CAR lymphocytes, as they were shown to normalize the IL-6 level and did not interfere with their antitumor action. Secondly, it is intriguing that FucCS does not inhibit lymphocyte activation and further gene modification but even promotes their transfection in our hands. This unexpected and currently unexplained advantage of FucCS addition during lymphocyte activation will be addressed in our future experiments. We hope that the addition of such substances at the CAR lymphocyte production step might be favorable for further therapy. However, additional experiments are necessary to find out the FucCS influence on cytokine production by CAR lymphocytes and their antitumor potential in vivo.

## 5. Patents

Patent of Russian Federation No. 2728361 29 July 2020 “Biomedical cellular product with HER2-specific anti-cancer activity” Kiselevsky M.V., Petkevich A.A., Chikileva I.O., Anisimova N.Yu.

## Figures and Tables

**Figure 1 biomedicines-11-02563-f001:**
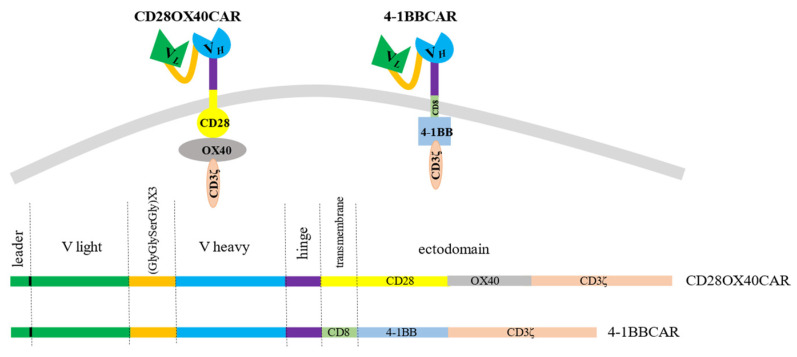
The 3rd generation OX-40-CD28-CAR with OX-40 signaling moiety added between CD28 and CD3ζ domains and the 2nd generation 4-1BB-CAR with 4-1BB signaling domain between CD8 transmembrane and CD3ζ signaling moieties.

**Figure 2 biomedicines-11-02563-f002:**
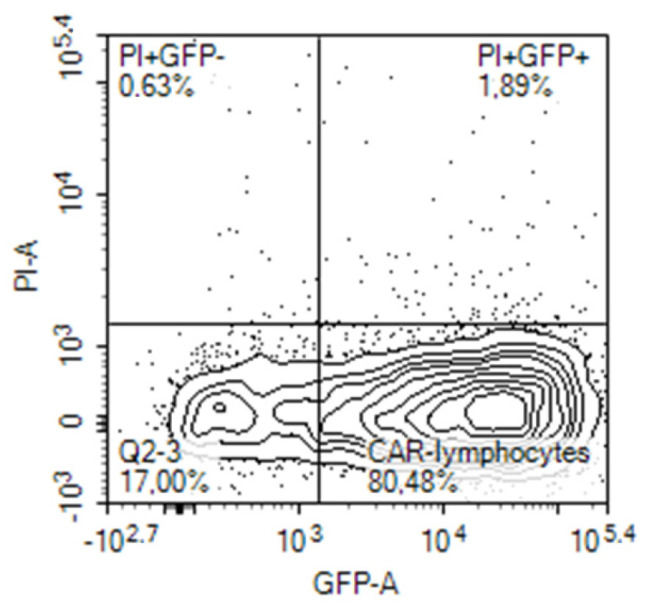
GFP expression by activated human PBLs that were transfected by nucleofection with CD28-OX40-CAR-IRES-GFP construct.

**Figure 3 biomedicines-11-02563-f003:**
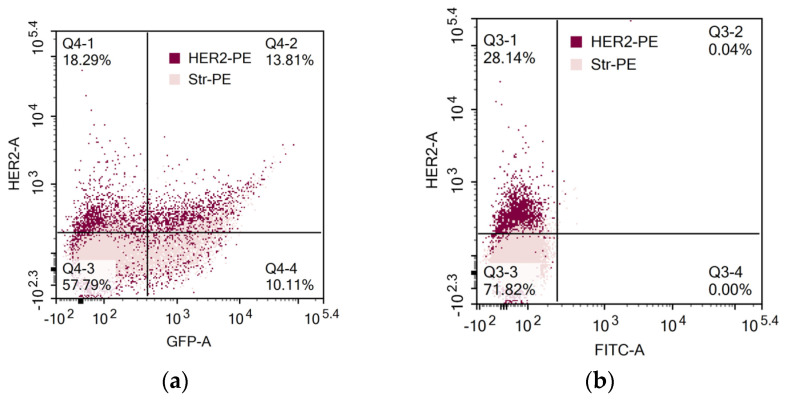
Evaluation of CAR expression. (**a**) GFP and CAR expression in activated human PBLs that were transfected by nucleofection with CD28-OX40-CAR-IRES-GFP construct. (**b**) CAR expression in 4-1B-B CAR lymphocytes. CAR expression was determined by sequential labeling of the cells with biotin-modified recombinant human HER2 and streptavidin-PE.

**Figure 4 biomedicines-11-02563-f004:**
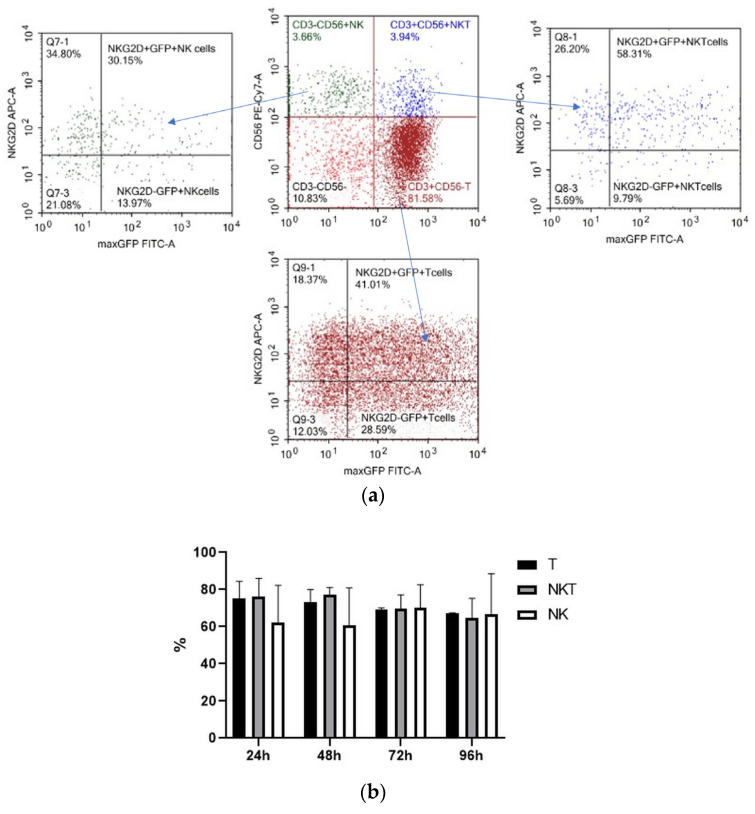
Expression of GFP, CD3, CD56, and NKG2D in pMax-GFP-electroporated lymphocytes during the culture period. (**a**) NK, NKT, and T cells were gated in lymphocyte population based on CD3 and CD56 expression. Therefore, GFP and NKG2D expressions were evaluated within the NK, NKT, and T subpopulations. Representative dot-plots of one of the donors at 24 h of incubation. (**b**) Ratio of GFP positive cells within lymphocyte subtypes at different time points. Values are the average of two donors and are expressed as mean ± SD.

**Figure 5 biomedicines-11-02563-f005:**
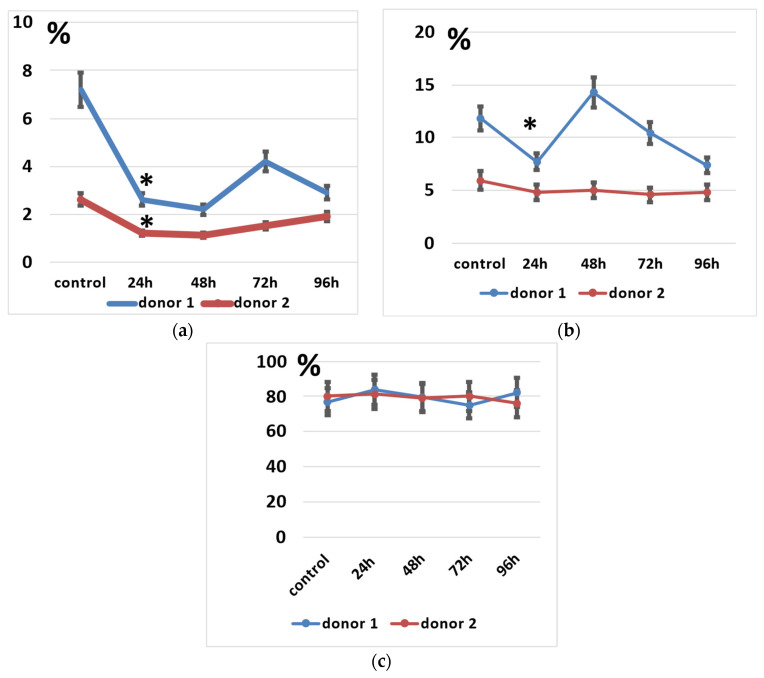
Lymphocyte subpopulations after the electroporation procedure. (**a**) NK cell subpopulation percentages during incubation. (**b**) NKT cell subpopulation percentages during incubation. (**c**) T cell subpopulation percentages during incubation. Data of two donors are presented as mean ± SD of triplicated evaluations. * statistically significant difference compared with control non-transfected cells at *p* ≤ 0.05.

**Figure 6 biomedicines-11-02563-f006:**
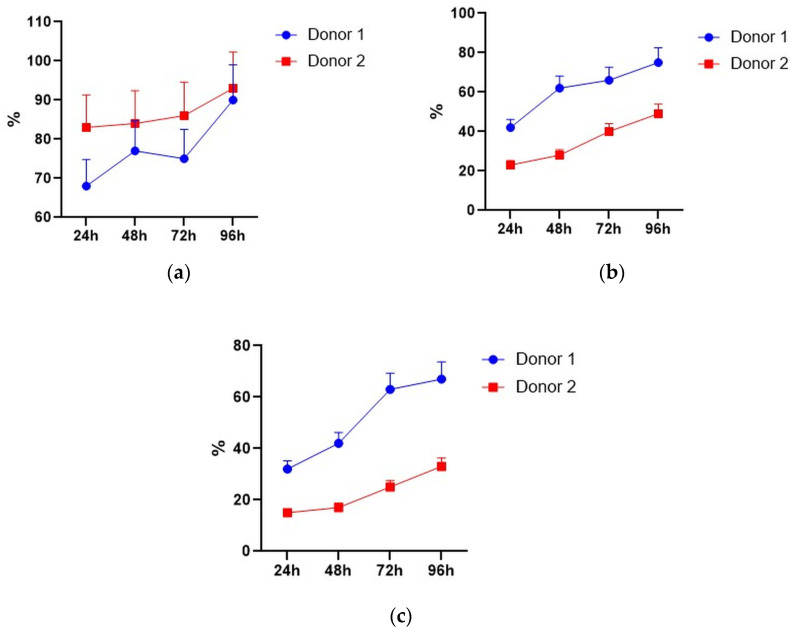
NKG2D-positive cells within GFP-positive NK, NKT, and T cells. (**a**) NK cells. (**b**) NKT cells. (**c**) T cells. Data of two donors are presented as mean ± SD of triplicated evaluations.

**Figure 7 biomedicines-11-02563-f007:**
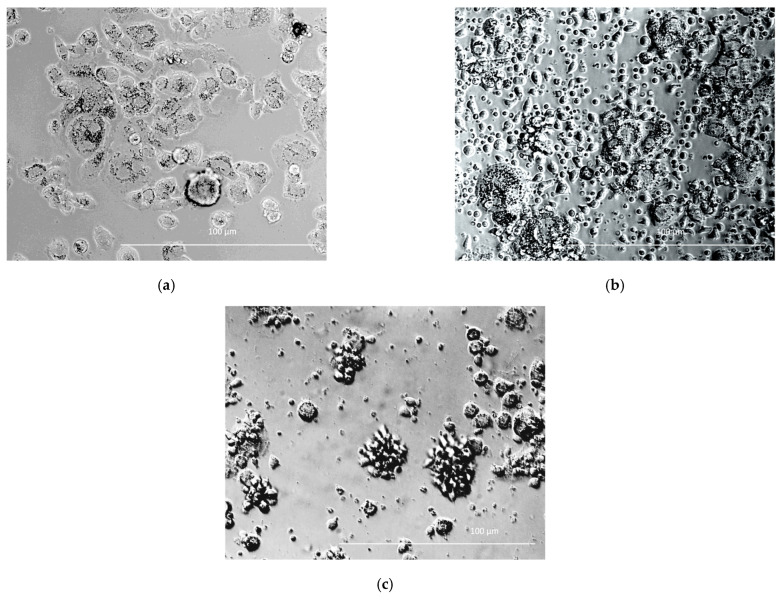
SKBR3 cells cultured alone (**a**) and co-cultured either with activated pMax-GFP-transfected PBLs (**b**) or CD28-OX40-CAR-T-NK cells (**c**). Phase-contrast microscopy, magnification eyepiece magnification, 10×; lens magnification, 40×.

**Figure 8 biomedicines-11-02563-f008:**
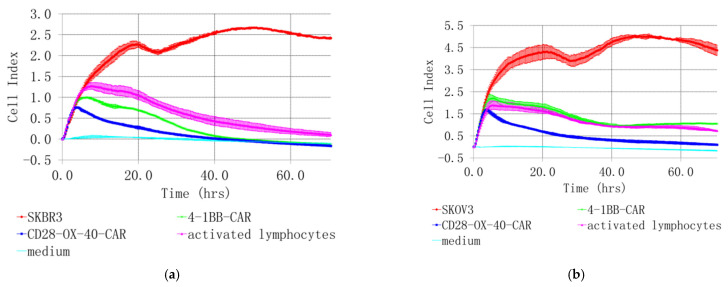
Real-time analysis of CD28-OX-40- and 4-1BB- HER2-specific CAR-T/NK cell cytotoxicity towards SKBR3 breast (**a**) and SKOV3 (**b**) ovarian cancer cells with XCELLigence instrument. For real-time analysis of the CAR-T/NK cell cytotoxicity towards cancer cells with XCELLigence instrument, 5 × 103 SKBR3 or SKOV3 cells were seeded into the strips and cultured for 4 h. Afterwards, 5 × 10^4^ (10× excess) of the CAR T/NK cells or activated T/NK cells were added. The data were collected every 15 min for 72 h. (**a**) H3,H4—medium control; A3,A4—control intact tumor cells SKBR3; G3,G4—SKBR3 plus control activated T/NK cells; C3,C4—SKBR3 plus 4-1BB-HER2-specific CAR-T/NK cells; D3,D4—SKBR3 plus CD28-OX-40-HER2-specific CAR-T/NK cells (**b**) H1,H2—medium control; A1,A2—control intact tumor cells SKOV3; G3,G4—SKOV3 plus control activated T/NK cells; C1,C2—SKOV3 plus 4-1BB-HER2-specific CAR-T/NK cells; D1,D2—SKOV3 plus CD28-OX-40-HER2-specific CAR-T/NK cells. Standard deviation (SD) is presented.

**Figure 9 biomedicines-11-02563-f009:**
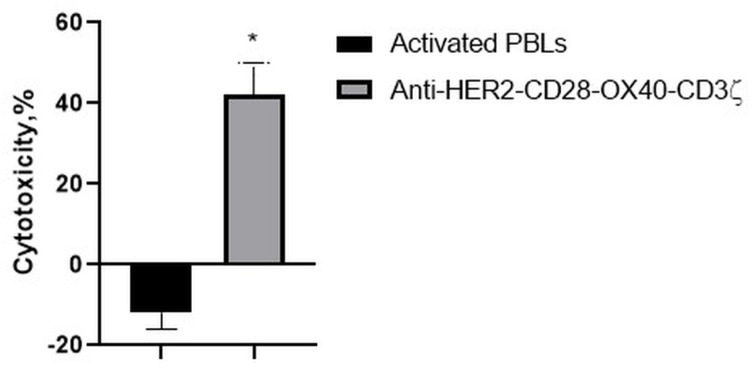
Cytotoxic action of the CD28-OX40-CAR-T/NK cells against a melanoma MTP tumor cell line. Data of one of 3 representative experiments are presented. Statistically significant difference compared with MTP cells co-incubated with activated PBLs * *p* ≤ 0.5.

**Figure 10 biomedicines-11-02563-f010:**
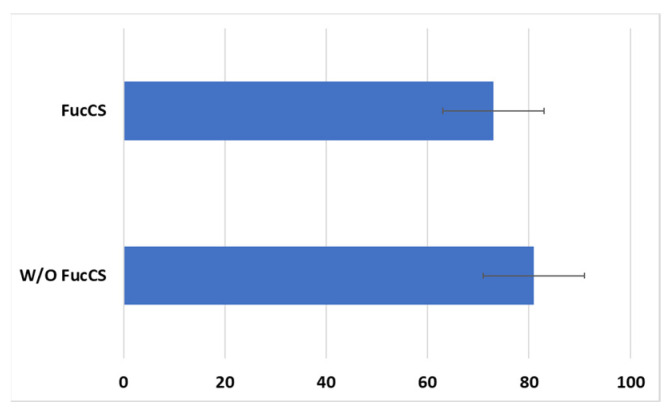
Cytotoxic action of the CD28-OX40-CAR-T/NK cells against SKBR3 tumor cell line in the presence or without FucCS. MTT test. Ratio of the effector cells towards tumor cells 5:1. Data of 1 of 3 representative experiments are presented.

**Figure 11 biomedicines-11-02563-f011:**
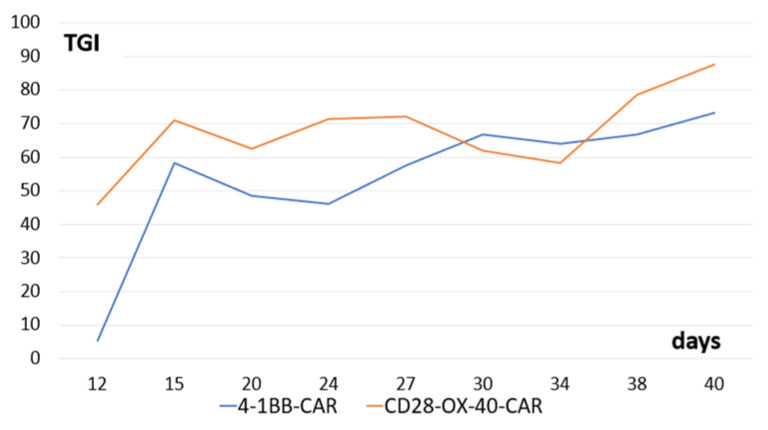
Tumor-growth inhibition compared with the untreated control (TGI) by CD28-OX-40-CAR—and 4-1BB-CAR-T/NK cells in mice with xenografts of human ovarian cancer SKOV3. Equal quantities (10^7^) of the tumor cells and the CAR-T/NK cells were injected subcutaneously into flanks of female nude mice. The control group of animals received only tumor cells. Tumor dimensions were measured with a digital caliper.

**Figure 12 biomedicines-11-02563-f012:**
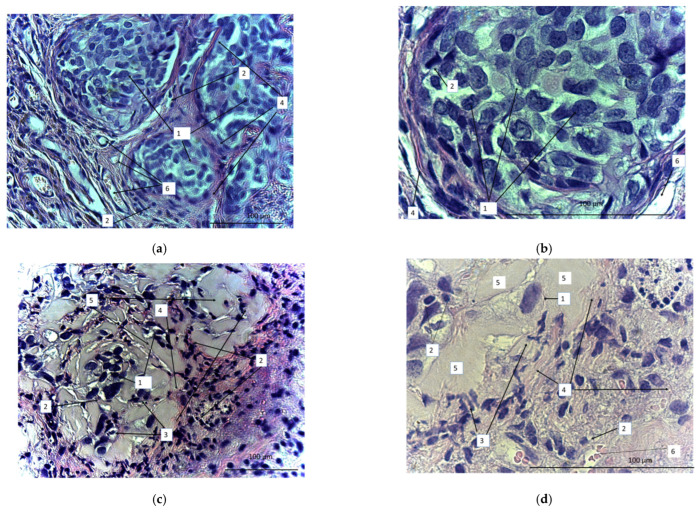
Micrographs of the sectioned SKOV3 xenograft tumors from either untreated (**a**,**b**) or CD28-OX-40-CAR-T/NK-treated nude mice (**c**,**d**). Magnification: 400 (**a**,**c**), 900 (**b**,**d**). 1: SKOV3 tumor cells; 2: lymphocytes; 3: apoptotic bodies; 4: stroma; 5: extracellular matrix; 6: blood vessels.

**Table 1 biomedicines-11-02563-t001:** Relative intensity of fluorescent signal after RONC-aH2-antibody-staining of several tumor cell lines.

Cell Line	Relative Interaction
SKBR3	+++
MCF7	++
SKOV3	+++
MTP	+
HELA	-
K562	-

-—negative; +—low expression; ++—moderate expression; +++—highly positive.

## Data Availability

The data presented in this study are available on request from the corresponding author. The data are not publicly available due to privacy restrictions.

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
