# Peer review of "Anti-Cancer Potential of Transiently Transfected HER2-Specific Human Mixed CAR-T and NK Cell Populations in Experimental Models: Initial Studies on Fucosylated Chondroitin Sulfate Usage for Safer Treatment"

_biomedicines, 2023, doi:10.3390/biomedicines11092563_

Round 1
Reviewer 1 Report
In this paper, you have presented interesting results concerning the use of antibody against HER-2 cloned in a plasmid vector containing sequences that increase the killing effect and then transiently transfect T and NK cells in the presence in some experiments of fucosylated chondroitin sulfate. Experiments performed were well designed, described in detail, well presented and analyzed, and thoroughly discussed. I understand that CAR immunotherapy is a hot topic and it has many advantages and restrains but yours in vivo experiments suggest that in vitro positive results could be applied also with promising results. Fucosylated chondroitin sulfate also appears to be beneficial in CAR immunotherapy of cancer.
Some corrections (suggestions)
Line 542. basing on àbased on
Line 527. Contrastingly à In contrast
Line 545, Strikingly à Furthermore
Line 166, 174, 465. dampen à reduce
Author Response
We appreciate all comments made by Reviewer 1 and provide below one-by-one responses.
Reviewer 1 Line 542. basing on à based on
Authors. Corrections were made. Please see the revised manuscript. Corrections highlighted in red.
Reviewer 1 Line 527. Contrastingly à In contrast
Authors. Corrections were made. Please see the revised manuscript. Corrections highlighted in red.
Reviewer 1 Line 545, Strikingly à Furthermore
Authors. Corrections were made. Please see the revised manuscript. Corrections highlighted in red.
Reviewer 1 Line 166, 174, 465. dampen à reduce
Authors. Corrections were made. Please see the revised manuscript. Corrections highlighted in red.
Reviewer 2 Report
Following are my comments for the manuscript:
1) Abstract is overall well written; In line 31, Adverse side effects is a wrong word (adverse and side has similar meaning); In line 32, typo -- it should be "off" tumor"; In line 33, it should be CAR constructs and not constructions
2) In line 62, it should be "off" tumor"; In line 133-134 it should be CAR constructs and not constructions; Overall very nicely written introduction! Very impressive
3) Materials and method section is very well written
4) In figure 2 & 3, how many days post EP were the CAR construct expression evaluated? Figure 2 and 3 would be better if there are bar graphs shown with mean/median values of replicates; Figure 3 a and b have different gating strategies which is inappropriate (gating strategies should be similar)
5) In figure 4a, very few events are captured, also, populations are mixed in gates; better colour schemes should be used for the left most panel; Figure 4b is missing labels for Y-axis; Bar graphs can be better (use prism)
6) Figure 5 is missing labels for Y-axis
7) In figure 6, Y-axis is missing labels, any thoughts on donor variability figure 6c; please redraw variability in graphs with prism
8) Figure 7 should look better with quantification of images
9) In figure 8, plot %cytolysis at different time points for better representation; were cells 100% CAR+ going into assay? If not, were they normalized according to target cell numbers? What was the E:T ratio for the assay?
10) What was E:T ration in figure 9? Better graphs are required; Y-axis labels are missing
11) Figure 11 is missing Axis labels; Untreated control is missing in the figure for TGI data
12) Have authors run MSDs/ELISA on supernatants collected during in vitro cytotoxicity assays for cytokine analysis? CRS?
13) Figure 12 should be quantified
14) Line 641 should be In line with...
15) Overall, discussion is very well written; any thoughts on in vivo use of FucCsu upon CAR T treatment to combat CRS?
Author Response
Reviewer 2
We appreciate all comments made by Reviewer 2 and provide below one-by-one responses.
Reviewer 2: 1. Abstract is overall well written; In line 31, Adverse side effects is a wrong word (adverse and side has similar meaning); In line 32, typo -- it should be "off" tumor"; In line 33, it should be CAR constructs and not constructions
Authors: Corrections were made. Please see the revised manuscript. Corrections highlighted in red.
Reviewer 2: 2. In line 62, it should be "off" tumor"; In line 133-134 it should be CAR constructs and not constructions; Overall very nicely written introduction! Very impressive
Authors: Corrections were made. Please see the revised manuscript. Corrections highlighted in red.
Reviewer 2: 3. Materials and method section is very well written
Reviewer 2: 4. In figure 2 & 3, how many days post EP were the CAR construct expression evaluated? Figure 2 and 3 would be better if there are bar graphs shown with mean/median values of replicates; Figure 3 a and b have different gating strategies which is inappropriate (gating strategies should be similar)
Authors: Expression of the CAR construct in lymphocytes after electroporation was evaluated on the next day, 24 hours after electroporation by different means. The information was added to the paper (lines 450-451, highlighted in red). Quadrants in figure 3 were unified.
Reviewer 2: 5. In figure 4a, very few events are captured, also, populations are mixed in gates; better colour schemes should be used for the left most panel; Figure 4b is missing labels for Y-axis; Bar graphs can be better (use prism)
Authors: For the figure 4a 15000 events were captured in the lymphocyte gate that is standard for our practice. However, it is a good idea to gather more events for better evaluation of minor populations such as NK and NKT cells. Thank you for your advice. We will certainly do so in our future experiments. Low mixture of lymphocyte populations is unfortunately inevitable in this case due to their incomplete resolution. Probably optimized fluorescent dyes could solve this problem. Color for the left most panel was panel was changed to dark green. Figure 4b was plotted in GraphPadPrism V8.0.1. Please, see the revised article.
Reviewer 2: 6. Figure 5 is missing labels for Y-axis.
Authors: Actually labels for Y-axis are present, but probably are poorly seen. The label (%) was enlarged. Please, see the revised version.
Reviewer 2: 7. In figure 6, Y-axis is missing labels, any thoughts on donor variability figure 6c; please redraw variability in graphs with prism
Authors: We do not have a precise explanation on donor variability. However, in our experience, NKG2D expression may be upregulated during or after infection diseases. Figure 4b was redrawn in GraphPadPrism V8.0.1. Please, see the revised version.
Reviewer 2: 8. Figure 7 should look better with quantification of images.
Authors: Unfortunately we don’t understand what kind of quantification is required. The scale bars are present in all of the microphotographs. Do you mean that they are not clear enough?
Reviewer 2: 9. In figure 8, plot %cytolysis at different time points for better representation; were cells 100% CAR+ going into assay? If not, were they normalized according to target cell numbers? What was the E:T ratio for the assay?
Authors: Figure 8 is derived via a special XCelligence RTCA Software. It can’t plot % cytolysis. Presentation of data in Cell Index is widely used and accepted. We suppose it is the most objective representation of the data from XCelligence machine without any manipulation from a researcher. However, if it is strictly required, we may try to export the data to Excel to calculate and draw such a graph.
CAR-lymphocyte preparations were not 100% positive. Actually, only about 30% of the cells were CAR-positive. However, plasmid DNA was strictly normalized during electroporation. Variability in expression was not significant (28-32%). Thus, we didn’t normalize quantities of CAR-cells. E:T ratio was 10:1 (lines 351-353 Materials and methods), ten lymphocytes per a tumor cell. However, number of CAR-lymphocytes was about 3 per a tumor cell.
Reviewer 2: 10. What was E:T ration in figure 9? Better graphs are required; Y-axis labels are missing
Authors: E:T ratio was 10:1 as indicated in Materials and methods (lines 334-335) . Figure 9 was redrawn in GraphPadPrism V8.0.1. Please, see the revised version.
Reviewer 2: 11. Figure 11 is missing Axis labels; Untreated control is missing in the figure for TGI data
Authors: Axis labels were present. However, in the revised manuscript we enlarged them to make more clear.
TGI (tumor growth inhibition, %) is calculated via the formula: [(Vc - Vt )/Vc ] × 100, where Vc is the mean tumor size of the vehicle control group (untreated control group) and Vt is the mean tumor size of the treated group. (Materials and methods, line 387). Thus, there is no need to plot TGI value for untreated control. It is 0% for all of the time points.
Reviewer 2: 12. Have authors run MSDs/ELISA on supernatants collected during in vitro cytotoxicity assays for cytokine analysis? CRS?
Authors: No, we have not yet. However, it is one of our primary goals in further research.(lines 722-723).
Reviewer 2: 13. Figure 12 should be quantified
Authors: Unfortunately we don’t understand what kind of quantification is required. The scale bars are present on all of the microphotographs. Do you mean that they are not clear enough?
Reviewer 2: 14. Line 641 should be In line with...
Authors: Corrections were made. Please see the revised manuscript they are highlighted in red. Line 642.
Reviewer 2: 15. Overall, discussion is very well written; any thoughts on in vivo use of FucCsu upon CAR T treatment to combat CRS?
Authors: Our current opinion on potential FucCs usage in vivo were summarized in lines 712-723 of the Discussion.